# Structural and Mechanismic Studies of Lactophoricin Analog, Novel Antibacterial Peptide

**DOI:** 10.3390/ijms22073734

**Published:** 2021-04-02

**Authors:** Minseon Kim, Jinyoung Son, Yongae Kim

**Affiliations:** Department of Chemistry, Hankuk University of Foreign Studies, Yongin 17035, Korea; alstjs032@naver.com (M.K.); jinyung7720@naver.com (J.S.)

**Keywords:** antibacterial peptide, lactophoricin, NMR spectroscopy, structural studies, antibacterial mechanisms

## Abstract

Naturally derived antibacterial peptides exhibit excellent pharmacological action without the risk of resistance, suggesting a potential role as biologicals. Lactophoricin-I (LPcin-I), found in the proteose peptone component-3 (PP3; lactophorin) of bovine milk, is known to exhibit antibiotic activity against Gram-positive and Gram-negative bacteria. Accordingly, we derived a new antibacterial peptide and investigated its structure–function relationship. This study was initiated by designing antibacterial peptide analogs with better antibacterial activity, less cytotoxicity, and shorter amino acid sequences based on LPcin-I. The structural properties of antibacterial peptide analogs were investigated via spectroscopic analysis, and the antibacterial activity was confirmed by measurement of the minimal inhibitory concentration (MIC). The structure and mechanism of the antibacterial peptide analog in the cell membrane were also studied via solution-state nuclear magnetic resonance (NMR) and solid-state NMR spectroscopy. Through ^15^N one-dimensional and two-dimensional NMR experiments and ^31^P NMR experiments, we suggest the 3D morphology and antibacterial mechanism in the phospholipid bilayer of the LPcin analog. This study is expected to establish a system for the development of novel antibacterial peptides and to establish a theoretical basis for research into antibiotic substitutes.

## 1. Introduction

Antibiotics have been considered one of the most important discoveries in modern medicine, but the indiscriminate misuse and abuse of antibiotics has resulted in the emergence of multidrug-resistant bacteria or superbacteria, which can resist various drugs [1,2,3,4,5,6,7]. For instance, methicillin-resistant *Staphylococcus aureus* (MRSA) is resistant to not only methicillin but also aminoglycosides, macrolides, tetracycline chloramphenicol, and lincosamides [8]. Expanded-spectrum β-lactamase (ESBL)-producing organisms confer resistance to penicillin, oxyiminocephalosporins, and monobactams [9]. The emergence of these superbacteria causes infectious disease that cannot be treated with conventional antibiotics. As new antibiotic substitutes are needed to overcome antibiotic-resistant bacteria, research into natural antibacterial peptides is ongoing to develop new types of antibiotics to resolve the fundamental limitations of the current commercially available antibiotics and advance the antibiotic market [10,11,12]. More than 3000 antibacterial peptides have been isolated from various organisms, and these studies suggest that antibacterial peptides are an important component of the innate immune system in most organisms, including plants and animals [13]. Most antibacterial peptides, with the cationic property, can kill pathogens at concentrations that do not harm the host cell [14]. Naturally derived antibacterial peptides have potential as new drugs because of their excellent pharmacological action in the human body [15,16].

In this study, we developed a cationic peptide that adheres to the cell membrane of anionic pathogens and destroys the cell membrane. Cationic antibacterial peptides include α-helical, β-sheet, loop, and extended peptides [17,18]. In particular, linear α-helical peptides occur mainly in the absence of a structure in an aqueous solution and contact with the hydrophobic membrane via formation of an amphipathic helix. The α-helix structure of cationic antibacterial peptides shows antibacterial activity against both Gram-positive and Gram-negative bacteria in direct proportion to the content of the α-helix [19].

Proteose peptone is a protein derived from bovine milk. Proteose peptone component-3 (PP3; lactophorin) is a hydrophilic phosphoglycoprotein, which is affected by the physicochemical properties of proteose peptone and plays an important biological role in bovine milk [20]. Lactophoricin-I (LPcin-I) is a cationic antibacterial peptide comprising 23 amino acids and represents the C-terminus residues 113–135 of PP3 [21,22,23,24]. LPcin-I exhibits antibacterial activity against Gram-positive and Gram-negative bacteria, without causing hemolysis of erythrocytes in humans [24]. LPcin-II is a cationic peptide corresponding to the 119–135 region of PP3. LPcin-I and LPcin-II exhibit similar charge ratios and the same hydrophilic/hydrophobic components based on the helical wheel pattern [25,26,27]. Both peptides are cationic, and the secondary structure is an α-helix in the phospholipid layer; however, only LPcin-I penetrates the planar lipid bilayer, and LPcin-II has no antibacterial activity [27]. Therefore, studies investigating the structural differences between LPcin-I and II have been reported [28]. Subsequently, based on LPcin-I, 20 types of peptides with improved antibacterial properties were designed via peptide engineering, and peptides with better antibacterial activity were developed [29,30]. In this study, YK5, one of the LPcin analogs, is mainly described. YK5 was expressed in large quantities in bacteria via genetic recombination. High-purity isolation and purification techniques were optimized to generate antibacterial peptides, and their structure and characteristics were analyzed [30].

Several mechanisms of antibacterial peptides against bacteria have been suggested. Mechanisms that can kill microbes without causing damage to the membrane can be divided into several groups, such as inhibition of macromolecular synthesis, inhibition of metabolic/enzymatic function, and inhibition of cell wall/membrane formation [31]. On the other hand, mechanisms that can kill microbes while damaging the membrane can be divided into the barrel-stave model, carpet pore formation, and the toroidal model. Among them, many studies of the interaction of amphipathic antibacterial peptides with membranes have focused on either a direct mechanism of bacterial killing through membrane permeabilization or modes of action involving an intracellular target [9,32]. These models elucidate the mechanism of action and bactericidal activity of antibacterial peptides. Therefore, a study of the interaction between antibacterial peptides and phospholipid membranes provides insights into antibacterial activity. The interaction between antibacterial peptides and phospholipid membranes was investigated using various analytical methods. The secondary structure of the peptides bound to the lipid bilayer was measured via circular dichroism (CD), and solution/solid-state NMR spectroscopy was used to determine the structure and orientation of the LPcin analogs in the biologically appropriate lipid bilayer [28,29,30,33,34,35,36]. In particular, solid-state NMR was used to identify the tertiary structure and 3D topology as well as to elucidate the mechanism of action of the LPcin analogs in the cell membrane [28,35,36]. Since the sample is highly viscous and sticky, it is difficult to use a commercial solid-state NMR probe. Therefore, a ^1^H−^15^N solid-state NMR probe and ^1^H−^31^P solid-state NMR probe were designed and constructed for experimental use [37].

## 2. Results

### 2.1. Design and Antibacterial Activity Test of Various LPcin Analogs

LPcin analogs with shorter amino acid sequences, less toxicity, and better antibacterial activity than LPcin-I were designed. Since antibacterial peptides are known to carry a net charge of +2 or more, serine residues (15th residue of LPcin-I) in the hydrophilic moiety were substituted with lysine to increase the net charge of the peptide and enable selective binding to the bacterial surface. In addition, it is known that unnecessarily high amphipathic properties increase toxicity or decrease cell selectivity. Therefore, the leucine residue (13th residue of LPcin-I) was mutated to a polar lysine to minimize toxicity by disrupting the α-helical amphipathic structure.

Agar diffusion was performed to derive peptides with excellent antibacterial activity in Gram-negative and Gram-positive bacteria. In this case, peptides synthesized by solid-phase synthesis were used. The larger the inhibition zone diameter, the greater the inhibition of bacterial growth. The antibacterial activity of the 20 peptides based on LPcin-I was similar to or greater than that of LPcin-I (Table 1). Novel antibacterial peptides exhibiting effective activity against Gram-negative and Gram-positive bacteria were selected via repeated experiments. Among the selected LPcin analogs, YK5 showed better bacterial growth inhibitory effect than LPcin-I in both Gram-negative and Gram-positive bacteria.

### 2.2. Expression of Selected LPcin Analog

In order to elucidate the mechanism of antibacterial activity of the peptides, a large number of antibacterial peptides is required. Expression using *E. coli* rather than organic synthesis is superior in terms of the quantity and quality of peptides produced, and also economically. Therefore, a peptide sample was obtained by optimizing the expression and efficiency of purification of the selected peptides using modified *E. coli* and a new strain appropriate for the expression of the antibacterial peptide.

The resulting expression vector for the selected YK5 was transformed into *E. coli*, for host cell expression. The annealed DNA for the expression of YK5 was separated via electrophoresis on a 2% agarose gel and was successfully purified. A 12% Tris-tricine SDS-PAGE gel electrophoresis was carried out to select the host cell expression and monitor the band thickness of the target fusion protein induced in each host cell line expressed (BL21(DE3)pLysS, C41(DE3), C43(DE3)). In Figure 1a, lanes 1, 4, and 7 show the expression before induction, whereas lanes 2, 5, and 8 show the expression 3 h after induction, and lanes 3, 6, and 9 show the expression 16 h after induction. Expression occurred by selecting C43(DE3) for YK5.

### 2.3. Peptide Isolation and Purification

The highly expressed *E. coli* was obtained as a pellet via centrifugation, and the transformed *E. coli* was disrupted using a lysis buffer, reagents such as lysozyme, and ultrasonication. The fusion protein was dissolved using a denaturant, and the KSI fused peptide was isolated via Ni-NTA affinity chromatography using His-tag. Various reagents and salts included in the KSI fused peptide were removed via dialysis, and the target peptide was isolated from the fusion peptide via chemical cleavage using CNBr. Expression and purification were confirmed by 12% Tris-tricine PAGE (Figure 1a). Peptides isolated via chemical cleavage using CNBr appear in lane 6 of Figure 1b.

All three peptides were finally purified by HPLC. It was found that band 2 in Figure 2a represents the YK5 peptide. The bands of each HPLC chromatogram were confirmed by 12% Tris-tricine gel electrophoresis. The approximate molecular weight was confirmed by the position of the peptide on the gel, and the target antibacterial peptide was separated and purified from the obtained eluent.

### 2.4. Mass Spectrometry and CD Spectroscopy

The molecular weights were measured to determine whether the peptide expressed in the transformed *E. coli* and isolated and purified via various assays was consistent with the target peptide. The molecular weight of YK5 peptides, which were finally purified via HPLC, was accurately measured using MALDI-TOF mass spectrometry. Chemical cleavage of the peptide yielded a homoserine lactone and homoserine-free acid. As shown in Figure 2b, the observed molecular mass was YK5 in 2008.1 Da. The precise mass of the homoserine lactone was 2007.41 Da for YK5, suggesting that the peptide exists as a homoserine lactone. The absence of other impurities was confirmed, suggesting optimization of the purification. The amino acid sequence of the LPcin analog was confirmed by de novo sequencing using tandem mass spectrometry (MS/MS) (Figure 3) [30].

The secondary structure of YK5 in the cell membrane was identified via CD. Dodecylphosphocholine (DPC) micelles were used in a membrane-mimicking environment, and the secondary structures of the three peptides were investigated with and without DPC micelles. In the absence of DPC micelles, the peptide exhibited a negative band at 200 nm, suggesting a random coil conformation. When the structure of the peptide was an ideal α-helix, the CD spectrum showed a double-dip indicating minimum values at 208 nm and 222 nm. In the presence of DPC micelles, a negative band around 220–230 nm demonstrated α-helical conformation as shown in Figure 2c.

### 2.5. Antibacterial Assays

A disc agar diffusion experiment was conducted to compare the antibacterial activity of the purified YK5 with that of the conventional antibacterial peptide. As a result of measuring the size of the inhibition region, the expressed peptide showed similar antibacterial activity to the synthesized peptide. This implies that YK5 shows more effective antibacterial activity against LPcin-I (data not shown). The micro-broth dilution test was used to determine the minimum concentration of antibacterial peptides that inhibit the growth of microorganisms, and tests were conducted for five pathogens (*L. innocua*, *S. aureus*, *P. aeruginosa*, *S. typhimurium*, and *E. coli*). Figure 4 shows the minimal inhibitory concentration (MIC) value of YK5. It can be seen that YK5 has a significantly smaller MIC value compared with LPcin-I, which suggests that the designed antibacterial peptide YK5 has better antibacterial activity compared with LPcin-I (data for LPcin-I are not shown). Next, a mammalian cytotoxicity test was conducted using mammalian cell lines (Table 2). The results confirmed cell stability even under high concentrations (100 μM or more) of antibiotic peptides, suggesting that YK5 is selectively involved in killing bacteria, with limited toxicity against mammalian cells.

Based on the antibacterial activity of YK5, changes in the cell morphology of *S. aureus* adapted to YK5 were investigated via scanning electron microscopy (SEM). *S. aureus* cells in the control group are generally round in shape (Figure 5a). However, treatment with YK5 induces morphological changes in *S. aureus* including rough surfaces and deformities (Figure 5b,c).

### 2.6. Solution-State NMR Spectroscopy

Since each cross-peak in the ^1^H−^15^N 2D HSQC spectrum shows the interaction between proton and nitrogen in the peptide bond, the presence and number of amine groups can be determined to establish the number of amino acid residues (Figure 6). Cross-peaks appearing at 8.05 ppm and 7.05 ppm of the ^1^H chemical shift and 113 ppm of the ^15^N chemical shift indicated the presence of N-H bond in the arginine side chain, and the peak corresponding to the arginine side chain was detected in the spectrum of all antibacterial peptides. Cross-peaks at 10.7 ppm of the ^1^H chemical shift in YK5 are attributed to N-H binding of tryptophan. The ^1^H−^15^N 2D HMQC-NOESY spectrum shows the connectivity between neighboring N-H cross-peaks along the polypeptide chain, which facilitates the identification of the amino acid sequence [33]. The whole ^15^N-labeled ^1^H−^15^N 2D HSQC spectrum of the N-H bond in the peptide, the ^15^N-labeled ^1^H−^15^N 2D HSQC spectrum selectively for specific amino acids, and the ^1^H−^15^N 2D HMQC-NOESY spectrum of the whole ^15^N-labeled peptide sample were determined (data not shown). Accordingly, the peptide arrangement and the degree of purification are easily known. As shown in the Figure 6, the whole ^15^N-labeled ^1^H−^15^N 2D HSQC spectrum in the micelle environment facilitated the identification of each cross-peak’s structure.

### 2.7. Solid-State NMR Spectroscopy

#### 2.7.1. ^15^N NMR Spectroscopy

Solid-state NMR was used to investigate the three-dimensional structure and absolute orientation of the antibiotic peptide in the cell membrane. Since the sample is in the form of a sticky liquid, a ^1^H−^15^N double-resonance home-built solid-state NMR probe with a solenoidal coil was designed and constructed for the experiment (Figure 7d) [36]. The peptide location on the surface of the cell membrane or within the cell membrane can be determined via ^1^H−^15^N 1D cross-polarization (CP). The ^1^H−^15^N 1D CP spectrum of YK5 appeared in the up-field region more than 120 ppm, indicating the peptide location on the cell membrane surface (Figure 7a) [38]. The ^1^H−^15^N 2D SAMPI4 spectrum uses the polarity index at slant angle (PISA) wheel, which represents a characteristic circular pattern, to illustrate the peptide morphology and topology in the cell membrane. Figure 7b shows that the calculated PISA wheel pattern overlapped with the ^1^H−^15^N 2D SAMPI4 spectrum for YK5, suggesting that YK5 contained two helical segments slanted at 42° (dot shape) and 57° (diamond shape) with respect to their normal bilayer. Accordingly, it was confirmed that the antimicrobial peptides show an α-helix structure and are inclined to the cell membrane surface at a specific angle. In addition, the relatively low value of the ^1^H−^15^N dipolar coupling constant confirmed that the YK5 peptide was located on the cell membrane surface [29,34,39]. The YK5 topology based on PISA wheel pattern analysis and MD simulation is presented in Figure 7c.

#### 2.7.2. ^31^P NMR Spectroscopy

The mechanism of the antibacterial peptide was analyzed by comparing the cell membrane environments with and without the antibiotic peptide via ^31^P solid-state NMR. The study was conducted in a cell membrane environment using a bicelle consisting of three types of phospholipids (14-O-PC, DMPG, 6-O-PC). A ^1^H−^31^P double-resonance home-built solid-state NMR probe with a 5 mm scroll coil was designed and constructed to develop a model simulating the antibacterial peptide mechanism on the cell membrane (Figure 8d) [36]. The figure represents a ^31^P 1D CP solid-state NMR spectrum of the bicelle with different concentrations of YK5. As shown in Figure 8a, the bicelle lacking YK5 revealed 14-O-PC (−10.8 ppm) and DMPG (−7.2 ppm) in the ^31^P spectrum [37,40], which represents the typical spectral pattern of a bicelle. Addition of 0.5 mM or 1 mM of YK5 to the bicelle (Figure 8b,c) maintained the peak pattern of 14-O-PC, but the peaks of DMPG and 6-O-PC were altered representing micelles or small vesicles [37,41,42]. Thus, the surface of the cell membrane is broken or changed by the antibacterial peptide. Recent models of antibacterial peptide mechanisms include a toroidal pore model in which the fatty acid head group constituting the cell membrane retains its original structure with the antibacterial peptide while forming a channel, and the barrel-stave model with altered orientation of ^31^P. A change in the ^31^P NMR spectrum of the bicelle using YK5 indicates penetration of the bilayer, which was apparently caused by lipid reconstitution, based on one of these structural models.

## 3. Materials and Methods

### 3.1. Design of Advanced LPcin Analogs

Shorter antibiotic peptides with simple amino acid composition are better candidates for clinical and commercial development. Therefore, based on LPcin-I consisting of 23 amino acids, a new antibacterial peptide derivative was designed with a shorter peptide length, enhanced antibacterial properties, and low hemolytic properties. Approximately, 20 LPcin analogs were designed by reducing or replacing the amino acid sequence. Since tryptophan carries an amphoteric indoline side chain, its location at the hydrophilic and hydrophobic interfaces enables hydrophobic and electrostatic interactions between the peptide and the bacterial cell membrane [43]. In addition, the hydrophobic region was substituted with a hydrophilic amino acid in order to increase the net charge by substituting the amino acid of the peptide hydrophilic region with lysine, and to diminish the amphipathicity [43,44,45]. The designed peptides were synthesized by Peptron (95% purity; Daejeon, Korea), and tested for antibacterial activity.

### 3.2. Antibacterial Activity of Peptides

The antibacterial activity of each peptide was tested by disc agar diffusion and broth dilution with Gram-positive bacteria (*Listeria innocua* MC2 KCTC 3658 and *Staphylococcus aureus* ATCC 6538) and Gram-negative bacteria (*Pseudomonas aeruginosa* ACTC 27853, *Salmonella typhimurium* ATCC 19430 and *E. coli* KCTC 1682). Five strains were cultured overnight in Brain heart infusion (BHI) medium in both tests. The primary antibacterial activities of the peptides were confirmed via disc agar diffusion using a 3.7% BHI agar plate. The prepared bacterial cultures were added to a BHI agar plate and maintained at 37 °C for 30 min. Sterile paper discs (6 mm) were prepared, immersed in 20 µL of each peptide solution, and then added to the prepared plate. After incubation in 37 °C ovens overnight, the diameter of the inhibitory region around each disc was measured to confirm the antibacterial activity.

To measure the minimum inhibitory concentration (MIC), a micro-broth dilution experiment was performed using a 96-well microplate photometer (Thermo Multiskan FC; Thermo Fisher Scientific, Waltham, MA, USA), and the MIC was defined at the lowest peptide concentration. Each peptide was diluted, followed by the addition of 50 µL to the bacterial suspension and incubation overnight at 37 °C. All antibacterial activity tests were performed according to the Clinical and Laboratory Standards Institute (CLSI) guideline, “Method for testing antibacterial susceptibility to aerobic bacteria; Approved Standard–Ninth Edition”.

### 3.3. Expression of Selected Peptide

The oligonucleotide-encoding peptide was synthesized by Intergrated DNA Technologies (Coralville, IA, USA) and cloned into pET31b (+) expression vector (Novagen, Madison, WI, USA). Based on NMR studies, the ^15^N-enriched peptide was synthesized in M9 minimal medium using ^15^NH_4_Cl as a nitrogen source. A 10 mL aliquot of the culture solution grown in Luria-Bertani (LB) medium was transferred to M9 minimal medium and incubated at 37 °C until a value of 0.5–0.6 was reached at 600 nm optical density (OD_600_). Isopropyl β-d−1-thiogalactopyranoside (IPTG) was added to a final concentration of 1 mM to induce a recombinant fusion protein. After 16 h, the cells were harvested by centrifugation at 6000 rpm for 30 min at 4 °C and stored in a −80 °C freezer for effective cell lysis.

### 3.4. Isolation and Purification of Peptide

In order to purify the harvested cells, the frozen cell pellets were resuspended in a lysis buffer (20 mM Tris, 500 mM NaCl, 15% glycerol) containing 1 mg/mL lysozyme (Sigma, New York, NY, USA) for ultrasonication. Cell lysates were centrifuged at 13,200 rpm for 30 min to obtain a pellet, which was dissolved in an Ni-NTA binding buffer (20 mM Tris, 500 mM NaCl, 5 mM Imidzaole, pH 8.0) containing 6 M guanidine-HCl (Fluka, Ronkonkoma, New York, NY, USA) for 5 h at room temperature. After centrifugation at 13,200 rpm, 4 °C and 30 min to remove impurities, a transparent supernatant was added to the Ni-NTA resin (Novagen, Madison, WI, USA) column, followed by elution with elution buffer containing imidazole. Guanidine and salt were removed via dialysis of eluates at room temperature for 24 h using distilled water, followed by lyophilization to chemically cleave the fusion partner and target protein using 70% formic acid and cyanogen bromide (CNBr, Sigma, New York, NY, USA). Peptides were purified using preparative RP HPLC (Waters Delta 600, Milford, MA, USA) on a Delta Pak C18 column (Waters, Milford, MA, USA) in a linear gradient of acetonitrile (ACN, 5–75%) in 0.1% trifluoroacetic acid (TFA).

### 3.5. Mass Spectrometry and CD Spectroscopy

Mass analysis of the peptides was performed using an MALDI-TOF MS analyzer (AB Sciex, Framingham, MA, USA). The purified peptides were dissolved in 30% ACN/0.1% TFA, mixed with a saturated solution of α-cyano−4-hydroxycinnamic acid (Sigma) matrix, loaded on a sample plate, and dried.

The secondary structure and morphology of the peptides were analyzed with a J−815 CD spectrophotometer (Jasco, Oklahoma city, OK, USA). The CD spectra of the three peptides were obtained in a water micelle environment. First, the peptide spectrum was measured in a water environment at room temperature, and then in the presence of dodecylphosphocholine (DPC) at concentrations of 20, 40, 60, 80, and 100 μm. The average blank spectrum (with solvent alone) was subtracted from the peptide spectrum.

### 3.6. Cytotoxicity Test

Cytotoxicity to normal cells in the human body was measured. The cytotoxicity assay of the expressed and purified peptides was tested by the Drug Discovery Platform Technology Group (Korea). Cytotoxicity tests were conducted in normal cells derived from various animals and humans. All mammalian cell lines used for cytotoxicity analysis were obtained from American Type Culture Collection (ATCC), and mammalian cell lines were VERO, HEL−1, L929, NIH 3T3, and CHO-K1. Each cell in turn represents an African green monkey kidney cell line, human embryonic lung cell line, mouse fibroblast cell line, mouse embryonic fibroblast cell line, and Chinese hamster ovary cell line. These cells were cultured in a 96-well plate at a seeding density of 1 × 10^4^ cells per well and treated with LPcin-YK5 peptide at a concentration ranging from 0.001 to 10 μM for 24 h. Then, 10 μL of Wst−8 solution was added to each well, and the cells were incubated at 37 °C for 1–4 h, and after 2 h in an infinite M1000 Pro reader (Switzerland), absorbance was measured at a wavelength of 450 nm.

### 3.7. Antibacterial Test by SEM Preparation of DPC Micelle

Based on the strong antibacterial activity of YK5 against *S. aureus*, the morphological changes in the bacteria were observed via SEM. *S. aureus* is a Gram-negative and round bacterium. A cell concentration of 1.5 × 10^7^ CFU/mL was used for EM images. Control without YK5 peptide, *S. aureus* with 20 µM (40 µg/mL) and 200 µM (400 µg/mL) of YK5, respectively, were compared to identify morphological changes in bacteria. *S. aureus* cells were treated with LPcin-YK5 at 37 °C for 3 h and then centrifuged at 5000× *g* for 5 min to obtain cells in the form of pellets. Thereafter, the strains were fixed in 2% paraformaldehyde (PFA) and 2% glutaraldehyde in 0.1 M phosphate buffer (pH 7.4, PB). The strains were postfixed with 1% osmium tetroxide dissolved in 0.1 M PB and dehydrated with gradually rising series (50–100%) of ethanol. Subsequently, isoamyl acetate was permeated and applied to a critical point dryer (HCP−2, Hitachi, Chiyoda, Tokyo, Japan), and the strains were coated with gold by ion sputter (IB−3 Eiko, Sumida, Tokyo, Japan). Samples were inspected and imaged with an analytical high resolution SEM (Carl Zeiss, Oberkochen, Germany) at the Korea Basic Science Research Institute Chuncheon Center (KBSI, Chuncheon, Gangwon, Korea).

### 3.8. Preparation of DPC Micelle

Purified ^15^N-labeled peptide was dissolved in 100 mM DPC-d_38_ micelles at pH 4.0 with 90% distilled deionized water (ddH_2_O) and 10% D_2_O, followed by centrifugation. The supernatant was collected into a 5 mm NMR sample tube. The final concentration of DPC-d_38_ and peptide was 100 mM and 1 mM.

### 3.9. Solution-State NMR Spectroscopy

All NMR spectra were measured using a Bruker Avance III HD spectrometer with an Ascend^TM^ 9.4 T standard bore magnet. The 2D HSQC (Heteronuclear Single Quantum Corrleation) and HMQC (Heteronuclear Multiple Quantum Correlation)–NOESY (Nuclear Overhauser Effect Spectroscopy) experiments were performed using an inverse probe at 315 K. All 2D NMR spectra were used with a complex point of 2048 and 64 transients per spectrum acquisition. A spectral window was used at 16 ppm in F2 and 40 ppm in F1 with relaxation delays of 2 s. The J_NH_ value was 90 Hz, and the mixing time was 200 ms. All NMR data were processed using Topspin software (Bruker).

### 3.10. Preparation of Phospholipid Bicelles

Purified ^15^N-labeled YK5 peptide (0.5 mg) was dissolved in 500 µL solution (450 µL of ddH_2_O and 50 µL of D_2_O) with perdeuterate DPC, and the final pH of micelle samples was 4.0. 1,2-di-*O*-tetradecyl-sn-glycero−3-phosphocholine (14-O-PC; Avanti Polar Lipids, Birmingham, AL, USA), 1,2-di-*O*-hexyl-sn-glycero−3-phosphocholine (6-O-PC; Avanti Polar Lipids, USA), and 1,2-dimyristoyl-sn-glycero−3-phospho-(1′- rac-glycerol) (DMPG; Avanti Polar Lipids, USA) were used to obtain large bicelles. ^15^N-labeled and purified peptide was dissolved using hydrated 14-O-PC, 6-O-PC and DMPG in 200 µL of sterilized distilled water. The 200 µL bicelles were transferred into a 5 mm flat-bottom glass tube (New Era Enterprises, Vineland, NJ, USA) and blocked from air inflow using a rubber cap. Parafilm and Teflon tape were used to seal the flat-bottom glass tube tightly.

### 3.11. Solid-State NMR Spectroscopy

#### 3.11.1. ^15^N-NMR Spectroscopy

All NMR spectra were measured using a Bruker Avance III HD spectrometer with an Ascend^TM^ 9.4 T standard bore magnet. All experiments were performed using a ^1^H−^15^N double resonance home-built solid-state NMR probe with a solenoid coil in our lab. The 1D CP experiments were performed with a mixing time of 5 ms, 512 complex points, 10,240 transients and recycle delay of 5 s. The 2D NMR spectra were obtained using a SAMPI4 (Selective Averaging via Magic sandwich Pulses using π/4 flip pulses) pulse sequence, which is a further advanced pulse sequence than SAMMY (sandwich-based separated local field spectroscopy) or PISEMA (Polarization Inversion Spin Exchange at the Magic Angle) [46]. To obtain the 2D SAMPI4 spectrum, 1024 complex points were used in the F2 dimension, and the number of transients was 892 per spectrum acquisition. Ammonium sulfate at 26.8 ppm in the ^1^H−^15^N CP spectrum was used as a standard external chemical shift reference.

#### 3.11.2. PISA Wheel Pattern Analysis

The tilt angle τ of the peptides was determined by fitting the simulated PISA wheel pattern to the 2D ^1^H−^15^N SAMPI4 spectrum. Using MATLAB & SIMULINK ver. R2010a (The MathWorks, Inc., Natick, MA, USA), the PISA wheel patterns were calculated. The standard order parameter S = 0.85 and the dihedral angles of Φ = −68° and Ψ = −44° for YK5 were used for the fitting simulation. The principal values of the chemical shift tensor were: σ_11_ = 64 ppm, σ_22_ = 77 ppm, and σ_33_ = 222 ppm for ^15^N and σ_11_ = 3 ppm, σ_22_ = 8 ppm, and σ_33_ = 17 ppm for ^1^H [40].

#### 3.11.3. Molecular Simulations

The simulations for three peptides were conducted using the CHARMM−27 (c35b5) all-atom force field [47,48] in Discovery Studio (DS) 2016 (Accelrys, San Diego, CA, USA). The SHAKE constraint was used in the simulations to maintain the position of the hydrogen bond. The “build and edit protocols” in DS were used to obtain the initial structures of the three peptides and determine the initial geometry before the simulation was refined via an energy minimization protocol involving the steepest descent. The location of the initial structure was altered along the bilayer normal axis to ensure a separation of 0.5 Å (z = −11.5–11.5 Å; z = 0 is the center of the bilayer normal axis). The thickness of the hydrophobic component of the membrane was 23 Å. An implicit membrane generalized Born model with a simple switching function (GBSW) was used for computational simulations given the interactions between the peptides and the model membrane [49]. The peptide simulations were carried out using the “Standard Dynamics Cascade” protocol with the leap-frog Verlet dynamics integrator and canonical NVT (number of particles, volume, and temperature in the isolated system) ensemble. This combination prevents alterations in the thermal equilibrium parameters during molecular simulation. Using the molecular simulation products from DS 2016 and the membrane builder module in CHARMM-GUI (http://www.charmm-gui.org/, accessed on 15 May 2018), real membrane models were designed to simulate phospholipid bilayer environments. All results were visualized by DS 2016 [28].

#### 3.11.4. ^31^P NMR Spectroscopy

The ^1^H−^31^P spectra were also measured using a Bruker Avance III HD spectrometer with an Ascend^TM^ 9.4 T standard bore magnet. A ^1^H−^31^P double resonance home-built solid-state NMR probe equipped with a scroll coil in our lab was used as the probe. The ^31^P 1D direct polarization (DP) spectra involved 10,240 complex points and 16 transients. The recycle delay between transients was 5 s, and the π/2 pulse length of ^31^P was 4 µs. A standard external chemical shift of 85% H_3_PO_4_ corresponding to 0 ppm in the ^31^P NMR spectrum was used as the reference.

## 4. Conclusions

The study of natural antibacterial peptides with excellent pharmacological activity and minimal resistance, and the development of novel biological products as antibiotic substitutes have increased. Novel peptides have been designed to enhance the antibacterial activity, with minimal cytotoxicity and short amino acid sequences based on LPcin-I, which is an antibacterial compound found in bovine milk. To investigate the interaction between the antibacterial peptides and the cell membrane, antibacterial peptides were produced at a large scale via expression, isolation, and purification of a recombinant protein. The antibacterial effect of the purified peptide was confirmed via various antibacterial assays. Subsequently, the tertiary structure and the antibacterial mechanism of the peptide in the cell membrane environment were analyzed using NMR spectroscopy. A ^1^H−^15^N home-built solid-state NMR probe with a solenoidal coil and a ^1^H−^31^P home-built solid-state NMR probe with a scroll coil were also designed and constructed. Based on the ^31^P NMR spectra, the orientation of the phospholipid component of the cell membrane was changed by the antibacterial peptide. As expected, the antibacterial peptide formed toroidal pores or channels in the cell membrane. Additional experiments in the future are necessary to reveal the correlation between these antibacterial peptides and phospholipid bilayers. Development of novel antibacterial peptides as described in this study provides a theoretical basis for investigation into antibiotic substitutes and lays the foundation for industrial manufacture of novel peptide drugs.

## Figures and Tables

**Figure 1 ijms-22-03734-f001:**
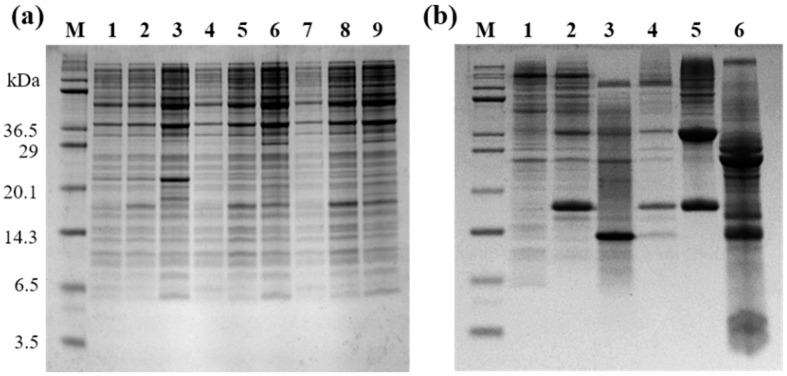
(**a**) Identification of host cells expressing a single insert of YK5 using 12% Tris-tricine PAGE. Lane M: molecular weight marker; Lanes 1–3: BL21(DE3)pLysS; Lane 4–6: C41(DE3); Lanes 7–9: C43(DE3). (**b**) Confirmation of the expression and purification of LPcin analog using 12% Tris-tricine PAGE. Lane M: molecular weight marker; Lane 1: before induction; Lane 2: after induction; Lane 3: supernatant after cell lysis; Lane 4: pellet after cell lysis; Lane 5: after Ni-NTA affinity chromatography; Lane 6: after CNBr chemical cleavage.

**Figure 2 ijms-22-03734-f002:**
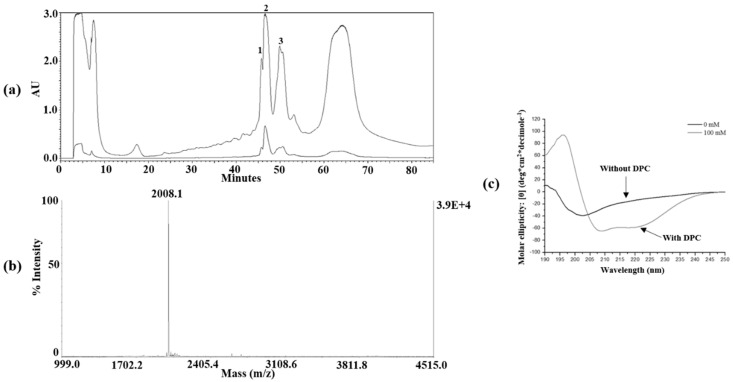
(**a**) HPLC chromatogram of YK5 and 12% Tris−tricine PAGE using reverse phase−C18 column. The band shown in lane 2 of 12% Tris−tricine PAGE is YK5. (**b**) MALDI−TOF mass spectrum of YK5. The spectrum was measured in positive ion reflector mode. (**c**) CD spectrum of YK5 showing a difference in secondary structure with or without dodecylphosphocholine (DPC) micelles.

**Figure 3 ijms-22-03734-f003:**
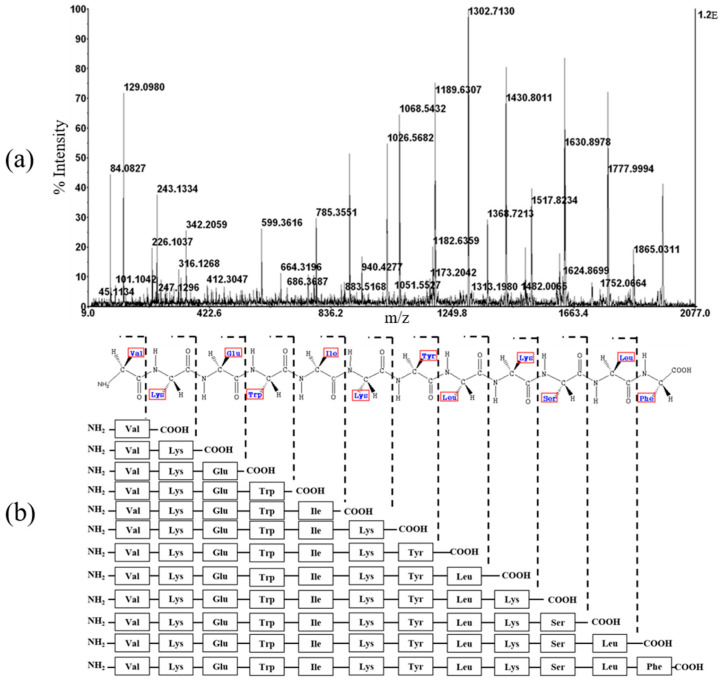
(**a**) De novo sequencing of YK3. (**b**) The m/z of the precursor ion was 1966.1 and using an automatic calculation method, the partial peptide sequence “VKEWIKYLKSLF” was determined.

**Figure 4 ijms-22-03734-f004:**
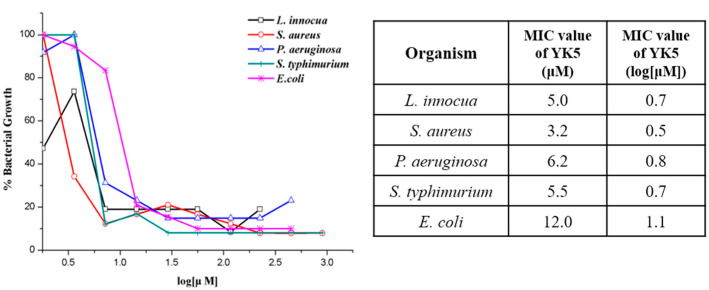
Micro-broth dilution tests and MIC values of YK5. (□) Listeria innocua, (○) Staphylococcus aureus, (△) Pseudomonas aeruginosa, (+) Salmonella typhimurium, and (*) Escherichia coli.

**Figure 5 ijms-22-03734-f005:**
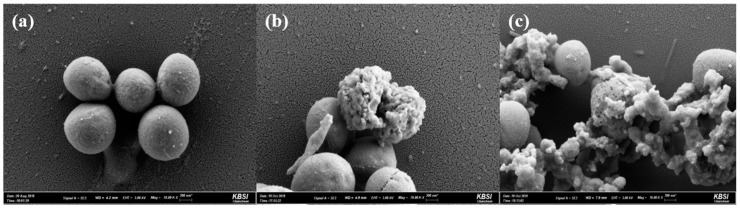
SEM photograph illustrating morphological changes of *S. aureus* in the absence of YK5 (**a**), and in the presence of 20 μM (**b**) and 200 μM of YK5 (**c**).

**Figure 6 ijms-22-03734-f006:**
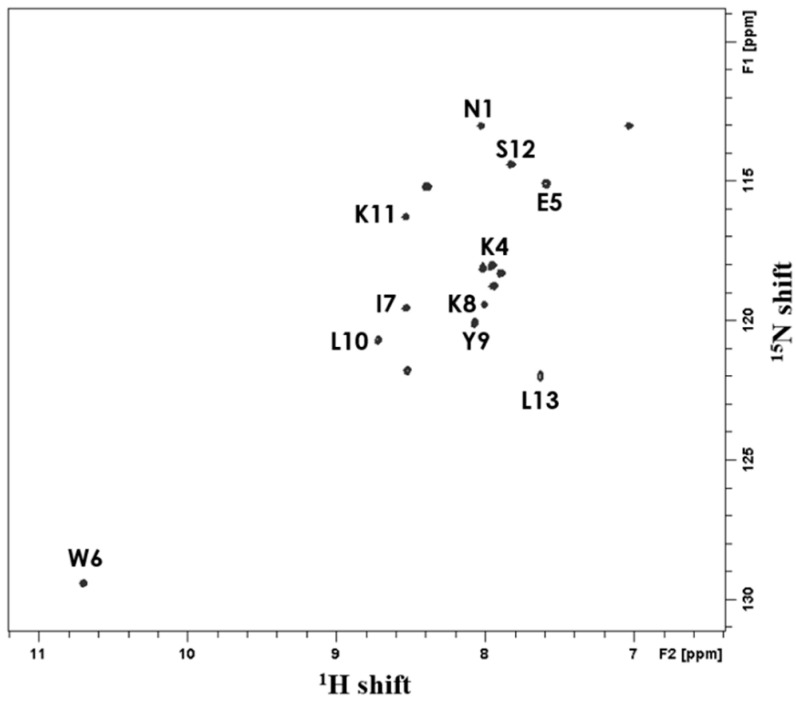
^1^H−^15^N HSQC spectrum of uniformly ^15^N labeled YK5 in DPC micelle. The ^1^H−^15^N cross-peaks were identified on the HSQC spectra of YK5 based on ^1^H−^15^N 2D HSQC spectra of selectively ^15^N-labeled peptides and 2D HMQC-NOESY spectra of uniformly ^15^N-labeled peptides.

**Figure 7 ijms-22-03734-f007:**
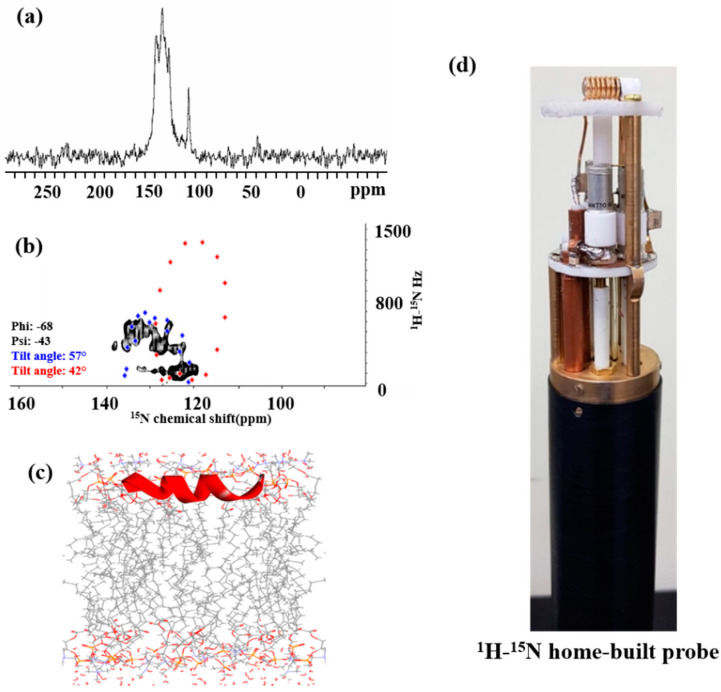
1D ^1^H−^15^N CP solid-state NMR spectrum (**a**) and 2D ^1^H−^15^N SAMPI4 spectrum overlapped with calculated polarity index at slant angle (PISA) wheel pattern of YK5 (**b**). (**c**) 3D topology of YK5 based on PISA wheel pattern analysis and molecular dynamics simulation. (**d**) ^1^H−^15^N home-built double-resonance solid-state NMR probe with a solenoidal coil.

**Figure 8 ijms-22-03734-f008:**
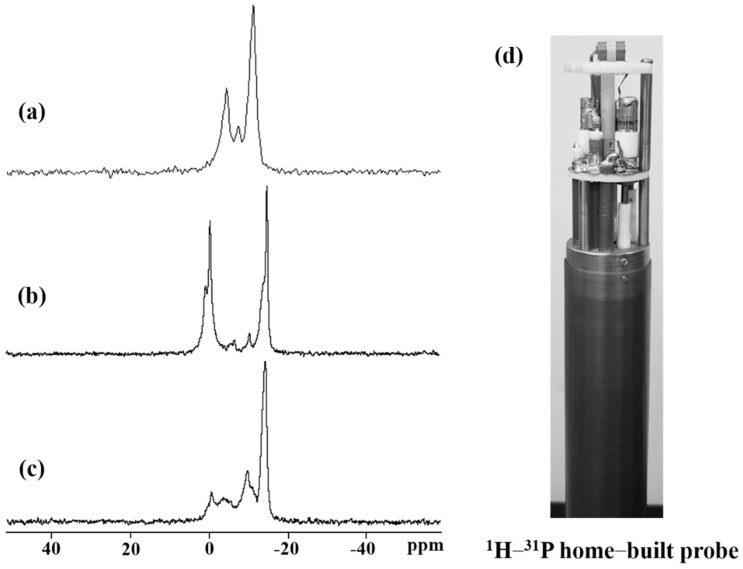
1D ^31^P solid−state NMR spectra of bicelle without YK5 (**a**), following the addition of 0.5 mM YK5 (**b**) and 1.0 mM YK5 (**c**) at 40 °C. (**d**) ^1^H−^15^N home-built double-resonance solid-state NMR probe with scroll coil.

**Table 1 ijms-22-03734-t001:** Amino acid sequences of Lactophoricin-I (LPcin-I) and its synthetic analogs.

Number	Peptide	Sequence	Net Charge at pH 7.0	Antimicrobial Activity	
				Gram-Positive	Gram-Negative
	LPcin-1	NTVKE TIKYL KSLFS HAFEV VKT	+ 2.1	+	++
1	LP1-C2	NTVKE TIKYL KSLFS HAFEV V	+ 1.1	–	+
2	LP1-C4	NTVKE TIKYL KSLFS HAFE	+ 1.1	+	++
3	LP1-C6	NTVKE TIKYL KSLFS HA	+ 2.1	+	+++
4	LP1-C8 (YK1)	NTVKE TIKYL KSLFS	+ 2.0	+	+++
5	LP1-C10	NTVKE TIKYL KSL	+ 2.0	–	–
6	LP1-T6W	NTVKE WIKYL KSLFS HAFEV VKT	+ 2.1	+	++
7	LP1-T2W	NWVKE TIKYL KSLFS HAFEV VKT	+ 2.1	++	++
8	LP1-T2,6W	NWVKE WIKYL KSLFS HAFEV VKT	+ 2.1	++	+++
9	LP1-T2K	NKVKE TIKYL KSLFS HAFEV VKT	+ 3.1	+	++
10	LP1-T2K,T6W (YK2)	NKVKE WIKYL KSLFS HAFEV VKT	+ 3.1	++	+++
11	LP1-T2K,T6W-C8 (YK3)	NKVKE WIKYL KSLFS	+ 3.0	++	+++
12	YK4	NKVKE WWKWL KSLFS	+ 3.0	++	–
13	YK5	NKVKE WIKYL KSLFK	+ 4.0	++	+++
14	YK6	NKVKE WIKYL KSKFS	+ 4.0	++	++
15	YK7	NKVKE WWKWL KSLFK	+ 4.0	++	+++
16	YK8	NKVKE WIKYL KSKFK	+ 5.0	++	+++
17	YK9	NKVKE WWKWL KSL	+ 3.0	++	++
18	YK10	NKVKE WIKYL KKL	+ 4.0	+	++
19	YK11	NKVKE WWKWL KKL	+ 4.0	++	++
20	YK12	NKVKE WWKWL K	+ 3.0	+	+

**Table 2 ijms-22-03734-t002:** Cytotoxicity test of YK5 against various mammalian cell lines.

	Cell Lines	VERO	NIH 3T3	L929	HFL-1	CHO-K1
Peptides (μΜ)						
YK5 IC_50_ (μM)		>100	>100	>100	>100	57.0

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
