# Peer review of "Structural and Mechanismic Studies of Lactophoricin Analog, Novel Antibacterial Peptide"

_ijms, 2021, doi:10.3390/ijms22073734_

Round 1

Reviewer 1 Report

General comment:

This manuscript, entitled “Structural and mechanismic studies of Lactophoricin analog, novel antibacterial peptide, derived from bovine milk,” authored by Kim et al., investigated the structure-function relationship of Lactophoricin-I (LPcin-I) derived new antibacterial peptide. This work has attempted to synthesize and analyze the structural properties of improved cationic antibacterial peptides using several spectroscopic techniques. In my opinion, this is a useful review and is suitable for publication in Int. J. Mol. Sci. after the authors have addressed the following comments and questions:

Specific comments:

  • How cationic peptides discriminate against bacterial cells and human cells (RBC etc)? Is their application suitable for humans against bacterial infection?
  • The role of glycans cannot be ignored to see peptide and membrane interaction.
  • In Table 1, other than YK5, at least 6 other peptides have similar antimicrobial activity. Are there any specific properties shown for YK5 to be best to work?
  • Please mention the designated number of residues to avoid any confusion – line 96, 97, 98, 99, 100 – serine to lysine or leucine to lysine
  • Line 123 – please mention figure number
  • Figure 1b – which band should be considered for peptides (lowermost in 6th lane)? Line 139 – what is the approximate molecular weight in the figure?
  • 2D HSQC may help confirm CD results in the presence and absence of DPC - in 100 mM DPC folded conformation may have a more scattered chemical shift signal compared to in the absence of DPC. Is there any technical issue not using this complementary technique to confirm this?

Author Response

Point 1: How cationic peptides discriminate against bacterial cells and human cells (RBC etc)? Is their application suitable for humans against bacterial infection?

Response 1: Cationic peptide has cationic properties that promote the preferential binding of peptides to the negatively charged bacterial cytoplasmic membrane instead of the zwitterionic membrane of mammalian cells.

Point 2: The role of glycans cannot be ignored to see peptide and membrane interaction.

Response 2: The role of peptidoglycan in bacterial cells plays attracting the cationic peptide.

Point 3: In Table 1, other than YK5, at least 6 other peptides have similar antimicrobial activity. Are there any specific properties shown for YK5 to be best to work?

Response 3: The antimicrobial activity of the designed analog peptides was confirmed using a standardized disc agar diffusion test. Of these, YK4, YK5, YK8, YK11 formed inhibition zones similar to YK3. We selected YK5 which has antimicrobial effect against all five bacterial cell.

Point 4: Please mention the designated number of residues to avoid any confusion – line 96, 97, 98, 99, 100 – serine to lysine or leucine to lysine.

Response 4: As you suggested, the number of residue is added in line 96, 97, 98, 99, 100.

Point 5: Line 123 – please mention figure number.

Response 5: As you suggested, the figure number is added.

Point 6: Figure 1b – which band should be considered for peptides (lowermost in 6th lane)? Line 139 – what is the approximate molecular weight in the figure?

Response 6: Yes, please consider the lowermost in 6th lane. In this figure, the approximate molecular weight is about 3 kDa.

Point 7: 2D HSQC may help confirm CD results in the presence and absence of DPC - in 100 mM DPC folded conformation may have a more scattered chemical shift signal compared to in the absence of DPC. Is there any technical issue not using this complementary technique to confirm this?

Response 7: We did NMR experiments in presence and absence of DPC. Residues exhibiting a wide range of chemical shifts provide evidence that the peptide adopted a stable conformation in micelle environments, but not in water. Please see the reference 35 for more details.

Reviewer 2 Report

The article focuses on antibacterial peptide derived from bovine milk. The manuscript is well prepared, and contains interesting results. However, I would like to indicate some issues, which should be explained/corrected:

1) Some microorganisms names are not in italic.

2) It is not clear, why these (but not others) peptides were synthetized.

3) An abstract could be supplemented with some exact values about MIC.

Author Response

Point 1: Some microorganisms names are not in italic.

Response 1: As you suggested, we revised it.

Point 2: It is not clear, why these (but not others) peptides were synthetized.

Response 2: Genetic recombination was performed through peptide engineering. Each peptide was synthesized by solid-phase synthesis to confirm its antimicrobial activity, and the disc agar diffusion test was performed to select the final three peptides having remarkable antimicrobial activity.

Point 3: An abstract could be supplemented with some exact values about MIC.

Response 3: As you suggested, we added it.